# Rectal Cancer following Local Excision of Rectal Adenomas with Low-Grade Dysplasia—A Multicenter Study

**DOI:** 10.3390/jcm12031032

**Published:** 2023-01-29

**Authors:** Yaron Rudnicki, Nir Horesh, Assaf Harbi, Barak Lubianiker, Eraan Green, Guy Raveh, Moran Slavin, Lior Segev, Haim Gilshtein, Muhammad Khalifa, Alexander Barenboim, Nir Wasserberg, Marat Khaikin, Hagit Tulchinsky, Nidal Issa, Daniel Duek, Shmuel Avital, Ian White

**Affiliations:** 1Meir Medical Center, Department of Surgery, Faculty of Medicine, Tel Aviv University, Kfar Saba 4428164, Israel; 2Sheba Medical Center, Department of General Surgery B and Organ Transplantation, Faculty of Medicine, Tel Aviv University, Ramat Gan 5265601, Israel; 3Rambam Health Care Campus, Department of General Surgery, Rappaport Faculty of Medicine, Technion-Israel Institute of Technology, Haifa 3109601, Israel; 4Rabin Medical Center-Hasharon Hospital, Department of Surgery, Faculty of Medicine, Tel Aviv University, Petach Tikva 49100, Israel; 5Tel Aviv Sourasky Medical Center, Department of Surgery, Faculty of Medicine, Tel Aviv University, Tel Aviv 69978, Israel; 6Rabin Medical Center-Beilinson Hospital, Department of Surgery, Faculty of Medicine, Tel Aviv University, Petach Tikva 4941492, Israel

**Keywords:** low-grade dysplasia, rectal cancer, rectal polyps, local excision, recurrence

## Abstract

**Purpose**: Rectal polyps with low-grade dysplasia (LGD) can be removed by local excision surgery (LE). It is unclear whether these lesions pose a higher risk for recurrence and cancer development and might warrant an early repeat rectal endoscopy. This study aims to assess the rectal cancer rate following local excision of LGD rectal lesions. **Methods**: A retrospective multicenter study including all patients that underwent LE for rectal polyps over a period of 11 years was conducted. Demographic, clinical, and surgical data of patients with LGD werecollected and analyzed. **Results**: Out of 274 patients that underwent LE of rectal lesions, 81 (30%) had a pathology of LGD. The mean patient age was 65 ± 11 years, and 52 (64%) were male. The mean distance from the anal verge was 7.2 ± 4.3 cm, and the average lesion was 3.2 ± 1.8 cm. Full thickness resection was achieved in 68 patients (84%), and four (5%) had involved margins for LGD. Nine patients (11%) had local recurrence and developed rectal cancer in an average time interval of 19.3 ± 14.5 months, with seven of them (78%) diagnosed less than two years after the initial LE. Seven of the nine patients were treated with another local excision, whilst one had a low anterior resection, and one was treated with radiation. The mean follow-up time was 25.3 ± 22.4 months. **Conclusions**: Locally resected rectal polyps with LGD may carry a significant risk of recurring and developing cancer within two years. This data suggests patients should have a closer surveillance protocol in place.

## 1. Introduction

A rectal polyp is considered a precursor for rectal cancer according to the adenoma-adenocarcinoma sequence. The timely removal of these lesions is considered essential for preventing advancement to malignancy in up to 90% of cases [1,2]. After successfully removing a polyp, it is recommended to continue surveillance at certain time intervals due to the increased risk for recurrence and advancement to malignancy [3]. The time interval for endoscopic surveillance is extrapolated from multiple studies trying to define the level of risk by features such as the number of polyps, the size, personal and family history of colorectal cancer (CRC), and the grade of dysplasia [4,5,6,7]. Guidelines for post-polypectomy surveillance after removing a dysplastic lesion are often ambiguous regarding the distinction between rectal and colonic polyps, the implications of resection margins with dysplasia, depth of resection, and whether the resection was performed endoscopically or surgically. Some post-polypectomy surveillance guidelines recommend an interval of three years for repeat colonoscopy after removal of a dysplastic lesion to detect early recurrence of adenomas or even CRC [8].

To date, most rectal lesions are removed endoscopically. However, distal rectal lesions; large or suspicious rectal lesions, including polyps larger than 1 cm; sessile adenomas; and early-stage rectal cancerous masses are better served by surgical removal. LE surgery can be performed with a standard transanal excision (TAE) or minimally invasive surgery (MIS). Local excision surgery allows for a full-thickness resection, negative resection margins, and the ability to close the defect in the rectal wall following resection [9]. The reported rate of LGD histology in LE rectal lesions varies from 7% to 51% [10,11]. As transanal MIS platforms became more and more common, the ability to remove larger masses transanally grew, which in turn led to a much higher number of LGD lesions being excised. It is unknown whether these LGD rectal lesions pose a higher risk of recurrence and development of rectal cancer.

This study aimed to assess the rate of rectal malignancy following LE of LGD lesions in the rectum.

## 2. Materials and Methods

A multicenter retrospective study following all patients with rectal lesions resected with a transanal local excision approach was conducted from October 2010 to March 2020 (11 years) in six academic medical centers in Israel. A subsequent analysis of patients with a final pathology of low-grade dysplasia was conducted. The data collected included the operative platforms used (a standard transanal excision (TAE), a transanal minimally invasive surgery (TAMIS), and a transanal endoscopic microsurgery (TEM)); demographics characteristics (age, gender, body mass index (BMI); co-morbidities; American Society of Anesthesiology (ASA) score); preoperative studies performed, including endoscopy with rectal lesion biopsy (rigid proctoscopy, Flexible Sigmoidoscopy, Colonoscopy); abdominal and pelvic CT; and endorectal ultrasound (ERUS) or pelvic magnetic resonance imaging (MRI), or both. 

Operative and postoperative data were collected, including operative approach, surgical findings, length of hospital stay, postoperative complications, morbidity, and mortality. The Clavien-Dindo classification of surgical complications score was used to classify postoperative complications [12]. Pathology reports were reviewed for histological characteristics such as size and resection margins. Out-patient visits and follow-up charts were reviewed for malignant recurrence and treatment after diagnosis of rectal malignancy. There was no standardized follow-up protocol, and various surveillance protocols were noted among the various centers. Polyps with any other pathology except low-grade dysplasia were excluded from the cohort.

Approval of the institutional review boards of all six participating centers was attained for the study (IRB 0179-20-MMC). All respective institutional review boards waived the need for individual informed consent by each patient for this retrospective study.

Statistical analyses were performed using EZR (Version 1.55) and R software (version 4.1.2) (Chugai Igakusha: Tokyo, Japan). Continuous data were expressed as mean and standard deviation when normally distributed or otherwise as the median and interquartile range (IQR). Student-t test or Mann–Whitney U test was used to analyze continuous variables. Categorical data were expressed as numbers and proportions and analyzed using Fisher exact or Chi-Square test. A *p*-value < 0.05 was considered significant. 

## 3. Results

During the study, 274 patients underwent transanal local excision of rectal tumors. The histological findings in the pathology reports of 81 patients (30%) were polyps with low-grade dysplasia and are the focus of this study. The other 193 patients had lesions with malignancy or high-grade dysplasia (Figure 1). The mean patient age at diagnosis was 65 ± 11 years, 52 patients (64%) were male, and the mean body mass index (BMI) was 26.6 ± 4.8 kg/m^2^. Patient demographics and preoperative data are detailed in Table 1. The mean and SD follow-up time was 25.3 ± 22.4 months. 

### 3.1. Preoperative Workup

All 81 patients with final pathology of LGD underwent a colonoscopy for surveillance or colorectal symptoms that showed a rectal polyp with a mean size of 3.3 ± 3.1 cm, at an average distance of 7 ± 3.5 cm from the anal verge. Endoscopy reports of the type of polyps seen were 40 (49%) sessile polyps and 13 (16%) pedunculated polyps, and 28 (35%) had missing endoscopic data. Although the final pathology was low-grade dysplasia for all cases, the initial preoperative biopsy reports were low-grade dysplasia for 49 patients (60%), 12 (15%) high-grade dysplasia, 3 (4%) well-differentiated adenocarcinoma, 2 (2%) moderately differentiated adenocarcinomas, and 15 (19%) had missing biopsy data. Thirty-seven of the 81 patients (46%) had imaging with endorectal ultrasound (ERUS), and eight of 81 (10%) had pelvic magnetic resonance imaging (MRI), seven of whom had also undergone a ERUS (Table 1).

### 3.2. Surgical Techniques and Operational Findings

The rectal lesions were resected using three main platforms. Twenty of eighty-one patients (25%) underwent a standard “open” transanal excision, 36 (44%) were operated on using the TAMIS platform, and 25 (31%) via the TEM platform. The mean distance of the lesions from the anal verge was 7.2 ± 4.3 cm, with a mean lesion size of 2.8 ± 1.6 cm. The depth of resection was a full thickness resection in 68 (84%) patients, partial thickness resection in 6 (7%), mucosal resection in 4 (5%), and a piecemeal resection in 3 (4%) patients. The rectal wall defect after resection was closed with a running suture in 49 (61%) cases, with interrupted sutures in 27 (33%), and was left open in 5 (6%). There were no intraoperative complications, and only one case (1%) had an added laparoscopy that ruled out intraabdominal penetration. 

### 3.3. Complication and Pathology Reports

The average length of hospital stay was 3.6 ± 1.8 days, and there were 12 (15%) cases with postoperative complications. Two cases of bleeding, six of wound infection or abscesses, two of cardiac or respiratory complications, and two of transient fecal incontinence. Only four (5%) cases were regarded as a Clavien-Dindo complication score of 3b or more. From the pathology reports, the mean diameter of the lesions was 3.2 ± 1.8 cm. Forty-seven lesions (58%) had a clear margin of over 3 mm; 18 (22%) had clear margins, but were under 3 mm; four (5%) had involved margins; and margin data was missing in 12 patients (15%). One patient (1%) with an involved margin underwent a repeat local excision attempt (Table 2).

### 3.4. Follow-Up, Recurrence, and Rectal Cancer Rate

Nine (11%) patients with LGD were found to have local intraluminal rectal cancer on follow-up. The mean follow-up was 25.3 ± 22.4 months. The average time interval from the first local excision to the diagnosis of the cancerous recurrence was 19.3 ± 14.5 months (range 5.2–54 months), with seven of nine (78%) of them diagnosed within less than two years from the initial LE. Reviewing the original reports of these nine cases showed that they all had an original low-grade dysplasia on the preoperative endoscopic biopsy and on final postoperative pathology. The average size of the original lesions was 3.5 cm (range 1.4–7 cm). Six patients had a clear margin of over 3 mm, and three had involved margins with low-grade dysplasia. The original resection depth was full-thickness resection in four patients, one had a partial thickness resection, one had a mucosal resection, and three had a piecemeal resection in more than one piece. The treatment modality chosen following rectal cancer diagnosis was a redo local excision in seven patients, one underwent a low anterior resection, and one got radiotherapy alone (Table 3).

## 4. Discussion

This study demonstrates a high risk (11%) for local recurrence and the development of rectal cancer in patients with low-grade dysplastic adenoma resected transanally from the rectum. Seven of the nine patients that developed cancer were diagnosed within two years of the original local excision of the pre-cancerous rectal lesion. Unlike high grade dysplastic rectal lesions that are suspected of cancer until proven otherwise and those with an involved dysplastic margin prompting re-excision, low-grade dysplastic lesions are not considered cancerous and might lead to a less stern approach and follow up. There is little data in the literature on the risk of these LGD patients and therefore at what intervals they should be endoscopically or clinically followed up with. Studies focusing on the risk of developing a possible malignant tumor seen on surveillance colonoscopies after three years showed that patients that had mild or mild/moderate dysplasia at the index polypectomy had a 3.5–5.5% risk of developing an advanced adenoma, not necessarily cancer, throughout the colon and rectum, mainly in the proximal colon [13,14]. It is important to state that these studies relate to endoscopic polypectomies and not specifically to surgically removed lesions, which are usually not endoscopically resectable. 

A meta-analysis by Saini et al. tried to quantify the risk factors for an advanced adenoma to be found during a three-year surveillance colonoscopy. He found that patients with ≥3 adenomas, a large adenoma (≥1 cm), or a high-grade dysplasia at the index polypectomy are at an increased risk for recurrence of advanced adenomas and therefore might benefit from close surveillance colonoscopies [15]. Most colonoscopy surveillance guidelines for patients with a history of resected adenomas stratify patients into low or high risk for recurrent advanced adenomas and cancer. The allocation to low or high risk is based mainly on the size, the number of adenomas, and having an advanced adenoma at the index colonoscopy. However, they do not draw any distinction between colonic and rectal polyps [5,7]. 

The definition of an advanced adenoma in the gastroenterology literature is usually an adenoma that is ≥1 cm, has villous histology, has a high-grade dysplasia feature, or even colorectal cancer in it [16,17,18]. Adenomas with low-grade dysplasia are not considered advanced adenomas and, presumably, do not place the patient in the high-risk group. Often, LGD adenomas are larger than 1 cm and then are considered high risk. Most guidelines do not take colonic vs. rectal origin of the polyp, or even positive dysplastic margins, into account for risk stratification. All patients that developed rectal cancer in this cohort had original adenomas that were larger than one centimeter and would have been classified as high risk for recurrence and, according to local recommendations, would have been advised to return for a repeat colonoscopy in three years. Having said that, the data in this study showed that almost all patients who developed rectal cancer after resection of the LGD lesion had done so in under two years from the index LE.

Unlike rectal polyps, large unresectable colonic polyps are usually referred to surgery for segmental colonic resection. By doing so, we not only gain a proper histological diagnosis of the polyp but also employ a preventive measure for recurrence in that segment of large bowel (as surgical resection also includes lymph node clearance). As transanal local excision surgeries found their place in treating rectal lesions, especially MIS platforms such as TEM and TAMIS, it brought about a plethora of new challenges in how to regard pre-cancerous rectal polyps. Unlike endoscopic polypectomy from the rectum, LE surgery allows for larger polyps, full-thickness resection, and a higher regard for margins [10,19]. LE of these “unresectable” polyps replaced anterior resection, which would have been a more oncologically complete surgical treatment. Therefore, as these large rectal polyps can be removed entirely with the rectum left in place, is the patient at a higher risk, which leads to the question: should these patients adhere to a different surveillance time interval? 

The US Multi-Society Task Force of Colorectal Cancer (MSTF), the American Society of Colon and Rectal Surgeons (ASCRS), and the National Comprehensive Cancer Network (NCCN) guidelines all recommend a repeat surveillance colonoscopy after removing a dysplastic lesion after three years, to detect and prevent CRC with no distinction between colon and rectum or modus of resection [20,21,22]. 

Our data suggest that large rectal polyps (≥1 cm) with LGD should be viewed in a different light, and it would be prudent to address them as highly suspicious lesions with a high risk for rectal cancer. Most of the polyps in the cohort were sessile polyps, some had a high suspicion of adenocarcinoma from the endoscopic biopsy, and most did not undergo specific pelvic imaging such as ERUS or pelvic MRI. From a technical point of view, not all lesions had a full-thickness resection, as five out of the nine patients that developed cancer had a partial thickness or piecemeal resection of the lesions, and six out of the nine patients that developed cancer had a transanal excision (TEA), a non-MIS technique, which is less common today. There were some postoperative complications, and not all lesions had clear dysplastic margins. These factors might account for seeding of free dysplastic cells or involved margins in the rectum that may explain the development of rectal cancer later in life.

From a pathology point of view, a lenient approach to non-cancerous lesions is prevalent, as they are not regarded as dangerous and, as such, lead to a more cursory pathological report [23]. The International Collaboration on Cancer Reporting (ICCR) recently published recommendations on pathology reporting of colorectal local excision specimens. There is little emphasis on grading dysplastic lesions and no specific recommendation for low-grade dysplasia [24]. In contrast to these findings, it is important to state that most of the diagnosed rectal cancer lesions were found to be early rectal cancer and were treated with another local excision. One patient underwent a low anterior resection, and one had radiotherapy. There was no distant disease found and no disease-related deaths. 

The study’s limitations revolve around its retrospective nature and its prolonged eleven-year period, during which time MIS LE has evolved and might have changed the approach to different lesions, as seen by the fact that more than half of recurrences occurred after TAE, with a non-full thickness resection and positive dysplastic margins but representing real-life practices. Although the multicentricity nature of the study is considered an advantage, it might allow for differences in patient selection and treatment choice that might have influenced the results. The relatively small sample size did not allow for a risk factors analysis. Adding to that, there was no active search for patients that recurred with only an adenoma (LGD or HGD) with no cancer, which might have augmented our findings’ strengths.

## 5. Conclusions

Locally resected rectal polyps with low-grade dysplasia may carry a significant risk of recurring and developing cancer within two years. A more meticulous approach to the preoperative assessment, a full-thickness resection, the resection margins, and the didactic pathological report might aid in reducing this occurrence. These data suggest that these patients should adhere to a closer surveillance protocol. Future studies should focus on assessing the optimal surveillance protocols and adherence for early detection of recurrence.

## Figures and Tables

**Figure 1 jcm-12-01032-f001:**
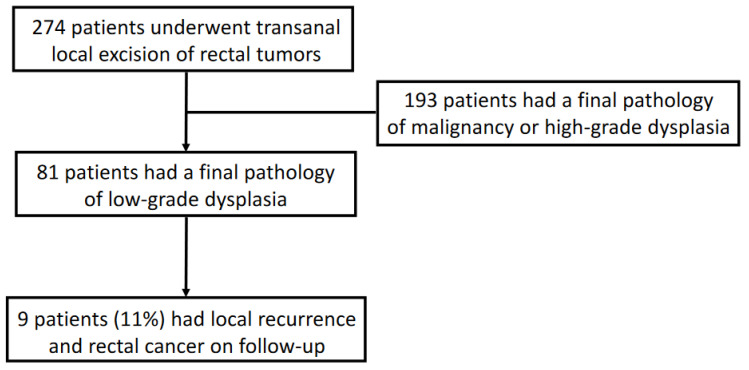
Flow chart of participation in the study, recurrence with rectal cancer.

**Table 1 jcm-12-01032-t001:** Baseline demographic and clinical and imaging characteristics of patients.

Characteristic	(*n* = 81)
Number of patients	81
Age at operation (years)—mean ± SD	65 ± 11
Male sex—*n* (%)	52 (64%)
Body mass index (BMI)—mean ± SD	26.6 ± 4.8
Co-morbidities ≥ 1 − *n* (%)	54 (67%)
ASA class I—*n* (%)	6
ASA class II—*n* (%)	47
ASA class III—*n* (%)	8
ASA class IV—*n* (%)	3
Missing data on ASA	17
**Preoperative assessment (Endoscopy)**	
Distance from anal verge (cm)—mean ± SD	7 ± 3.5
Largest diameter of lesion (cm)—mean ± SD	3.3 ± 3.1
Type of polyp	
Sessile polyp—*n* (%)	40 (49%)
Pedunculated polyp—*n* (%)	13 (16%)
Missing data—*n* (%)	28 (35%)
Preoperative biopsy report	
Low-grade dysplasia (LGD)	49 (60%)
High-grade dysplasia (HGD)	12 (15%)
Well-differentiated adenocarcinoma	3 (4%)
Moderately differentiated adenocarcinoma	2 (2%)
Poorly differentiated adenocarcinoma	0
Missing data	15 (19%)
Preoperative imaging	
ERUS was performed—*n* (%)	37 (46%)
MRI Pelvis was performed—*n* (%)	8 (10%)

ASA—American Society of Anesthesiology score; ERUS—Endorectal ultrasound; MRI—Magnetic resonance imaging.

**Table 2 jcm-12-01032-t002:** Surgical techniques, operational findings, complications, and pathology reports.

Characteristic	(*n* = 81)
Operative technique	
TAE—*n* (%)	20 (25%)
TAMIS—*n* (%)	36 (44%)
TEM—*n* (%)	25 (31%)
Distance from anal verge (cm)—mean ± SD	7.2 ± 4.3
Largest diameter of lesion (cm)—mean ± SD	2.8 ± 1.6
Predominant rectal wall location	
Posterior wall	19 (23%)
Anterior wall	18 (22%)
Lt lateral wall	21 (26%)
Rt lateral wall	13 (16%)
Unknown	10 (13%)
Depth of resection	
Full-thickness resection	68 (84%)
Partial-thickness resection	6 (7%)
Mucosal resection	4 (5%)
Piecemeal (in >1 piece)	3 (4%)
Defect closure approach	
Running suture	49 (61%)
Interrupted sutures	27 (33%)
Defect left open	5 (6%)
Intra-operative complications	0
Laparoscopy added—no complication found	1 (1%)
LOS—Length of stay (days)—mean ± SD	3.6 ± 1.8
Postoperative complications	12 (15%)
Bleeding	2 (2%)
Wound infection/Abscess	6 (7%)
Cardiac/Respiratory complication	2 (2%)
Transient fecal incontinence	2 (2%)
Clavien-Dindo ≥ 3B	4 (5%)
**Final pathology**	
Largest diameter of lesion (cm)—mean ± SD	3.2 ± 1.8
Margins	
Clear margins >3 mm	47 (58%)
Clear margins <3 mm	18 (22%)
Involved margins	4 (5%)
Missing data	12 (15%)
Added treatment after pathology—Redo LE	1 (1%)

TEA—Transanal excision; TAMIS—Transanal minimally invasive surgery; TEM—Transanal endoscopic microsurgery; Clavien-Dindo—The Clavien-Dindo classification of surgical complications; LE—Local excision.

**Table 3 jcm-12-01032-t003:** Characteristics of recurrence cases.

Characteristic	(*n* = 9)
Local recurrence	9
Systemic recurrence	0
Time interval from LE to cancerous recurrence (months)—mean ± SD (range)	19.3 ± 14.5 (5.2–54)
Number of patients that recurred under 24 months	7 (78%)
Original largest diameter of lesion (cm)—mean (range)	3.5 (1.5–7)
Original margins	
Clear margins >3 mm	6/9
Clear margins <3 mm	0
Involved margins	3/9
Original depth of resection	
Full-thickness resection	4/9
Partial-thickness resection	1/9
Mucosal resection	1/9
Piecemeal (in >1 piece)	3/9
Original operative platform used	
TAE—*p*/*n*	6/9
TAMIS—*p*/*n*	1/9
TEM—*p*/*n*	2/9
Treatment after recurrence	
Re-do local excision	7
LAR	1
Radiotherapy	1
Follow-up time (months)—mean ± SD	25.3 ± 22.4

LE—Local excision; SD—Standard deviation; *p*—Positive; *n*—Number; TEA—Transanal excision; TAMIS—Transanal minimally invasive surgery; TEM—Transanal endoscopic microsurgery; LAR—Low anterior resection.

## Data Availability

Data will be available upon request from authors.

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
