# Peer review of "Rectal Cancer following Local Excision of Rectal Adenomas with Low-Grade Dysplasia—A Multicenter Study"

_jcm, 2023, doi:10.3390/jcm12031032_

Round 1
Reviewer 1 Report
I have read with interest the work presented by the authors to describe the result of the retrospective and multicentre study analyzing the risk of developing rectal cancer after the resection of an adenoma with low-grade dysplasia.
It is a retrospective work but very well structured and shows that we are currently doing suboptimal monitoring of certain patients. As the authors explain, the current guidelines propose 3-yerar surveillance colonoscopy. But surgeons need to know that a rectal injury of more than 3 cm requires different follow-up. And not only because of the possible risk of recurrence as exposed in the work, but also because of the functional alteration that involves a resection of a bulky lesion of the rectum. As already demonstrated in different works.
The conclusions are justified with the results and what is exposed in the work. But I am of the opinion that surgeons who perform transanal surgery cannot act as technicians and once the infiltrative disease is ruled out, discharge patients according to the screening program.
Congratulating the authors for shedding light on a problem that currently, due to the pressure of reducing the number of colonoscopies in the follow-up, will come out in the coming years.
Author Response
We thank the reviewer for the support and kind words. It is not so often that reviewers focus on the positive parts of a manuscript, and we were very honored to get this response. We also believe this is an important paper that can shed some light on these specific patients and lesions and that surgeons can not act only as technicians once the infiltrative disease is ruled out.
Please see attached the revised manuscript.

Reviewer 2 Report
I must compliment Rudnicki et al on their manuscript, titled: “Rectal Cancer Following Local Excision of Rectal Adenomas With Low-Grade Dysplasia – A Multicenter Study”. The authors performed extensive data collection in 11 centers on patients that underwent local excision for rectal polyps. 81 out of 274 had low grade dysplasia, of which 11% developed rectal cancer within 2 years.
The manuscript is well written and it is easy to read. I enjoyed reviewing this manuscript. The analysis is clean and easy to understand. However, I have a few questions.
1. Do the authors have more details on the performed procedures?
2. What were the inclusion criteria for Tamis? Mean diameter of 3.2 seems rather small for surgical excision.
3. One obvious weak point is that this series is too small to perform a risk factor analysis. I would like to suggest mentioning this in the discussion section.
4. Surveillance protocol adherence seems like a logical conclusion and clinical recommendation. Do the authors have recommendations for future research as well?
Author Response
We thank the reviewer for the support and kind words and would like to address the remarks. Currently, the submission system does not allow for a revised manuscript re-submission since we are awaiting a report from another reviewer. Still, we can address the remarks here and send a revised full manuscript when the option opens up in the submission system.
- Other than the data reported in the manuscript, like lesion characteristics, the platform used (TAE, TEM, TAMIS), depth of resection, and intraoperative complications, we do not have more details on the performed procedures. We tried collecting information on what percentage of the circumference of the rectal wall was involved and were the lesions were mobile or fixed, but we could not find enough data on all patients.
- The inclusion criteria were mainly a rectal lesion unresectable endoscopically but also adhering to the NCCA guidelines for MIS local excision, such as < 30% circumference of the bowel, mobile, favorable histology, and located in the lower two-thirds of the rectum. The decision on TAMIS versus TEM or TAE was center and surgeon based. This cohort was a part of a larger cohort of LE of rectal lesions, and the mean diameter was larger in the group of malignant rectal lesions.
- In the revised manuscript, we add this remark to the limitation part of the discussion on page 8: "The relatively small sample size did not allow for a risk factors analysis."
- We add this recommendation in the conclusion part on page 8: "Future studies should focus on assessing the optimal surveillance protocols and adherence for early detection of recurrence."
